

# Single nucleotide polymorphism of *MTHFR* rs1801133 associated with elevated Hcy levels affects susceptibility to cerebral small vessel disease

Hongyu Yuan[1], Man Fu[2], Xianzhang Yang[1], Kun Huang[1] and Xiaoyan Ren[1]

[1] The Third Department of Neurology, Heze Municipal Hospital, Heze, Shandong Province, China
[2] Department of Neurology, The Third People's Hospital of Heze, Heze, Shandong Province, China

## ABSTRACT

**Background.** Methylenetetrahydrofolate reductase (MTHFR) is indispensable for the conversion of homocysteine (Hcy) to methionine. The single nucleotide polymorphism (SNP) of MTHFR gene (rs1801133, C667T) is correlated with decreased enzyme activity that eventually results in elevated plasma Hcy levels. Hyperhomocysteinemia has been confirmed to be involved in the pathogenesis of stroke, cerebral small vessel disease (CSVD), various metabolic disorders and so on. However, the relationship between the MTHFR gene polymorphisms, Hcy, and CSVD has not been investigated. In this study, the relationship between SNPs of MTHFR gene and CSVD was determined after adjusting for cardiovascular risk factors, and the potential mechanism based on Hcy levels was explored.

**Methods.** A total of 163 consecutive CSVD patients were collected as the case group. In the corresponding period, 326 healthy people were selected as the control group, who were matched to these cases according to age (±2 years) and gender at a ratio of 2:1. SNPs of MTHFR rs1801133, rs1801131, rs2274976, rs4846048, rs4846049, rs13306561 and rs3737964, were genotyped with TaqMan Pre-Designed SNP Genotyping Assays. Plasma Hcy levels were detected using Hcy reagent through enzymatic cycling assay. Multivariate analysis was used to identify the SNPs associated with CSVD susceptibility. Plasma Hcy levels were compared between different genotypes.

**Results.** The MTHFR rs1801133 TT and CT genotype had increased risk for CSVD, and the *OR* was higher in the TT genotype than in the CT genotype (2.307 vs 1.473). The plasma Hcy levels of different genotypes showed the tendency of the TT genotype > CT genotype > CC genotype (19.91 ± 8.73 pg/ml vs 17.04 ± 5.68 pg/ml vs 14.96 ± 4.85 pg/ml).

**Conclusions.** The SNP of MTHFR rs1801133 was correlated with CSVD, and the TT and CT genotypes had increased risk for CSVD compared to the CC genotype. The potential mechanism was associated with elevated Hcy levels.

Corresponding author
Xiaoyan Ren, ren_1667@sina.com

## INTRODUCTION

As a generic term for intracranial vascular disease associated with various neurological and pathological processes, cerebral small vessel disease (CSVD) refers to a syndrome of different clinical features and neuroimaging findings induced by pathological changes in capillaries, perforating cerebral arterioles and venules (*Shi & Wardlaw, 2016*). CSVD is the cause of 45% of all the cases of dementia, and accounts for 25% of the ischemic stroke cases around the world (*Pantoni, 2010*; *Petty et al., 2000*; *Wardlaw et al., 2013*). Additionally, CSVD is a major cause for depression, cognitive impairment, disability and so on in the aged (*Li et al., 2018*).

The pathogenesis for CSVD mainly includes impairment of cerebral autoregulation, reduction of cerebral blood flow and increase of blood–brain barrier (BBB) permeability (*Joutel & Chabriat, 2017*; *Li et al., 2018*; *Li et al., 2019*). However, its molecular mechanisms are not completely elucidated. Genetic studies have demonstrated that CSVD is highly heritable, especially in young-onset stroke patients, and that disease processes of some CSVD subtypes may be associated with common variants in monogenic disease genes (*Tan et al., 2017*).

Methylenetetrahydrofolate reductase (MTHFR) gene is located on chromosome 1p36.3, and it is associated with the biosynthesis pathway of amino acid and purine (*Goyette et al., 1994*; *Cui et al., 2011*). As an important regulatory enzyme catalyzing the transformation of 5, 10-methylenetetrahydrofolate to 5-methyltetrahydrofolate, MTHFR is indispensable for the conversion of homocysteine (Hcy) to methionine (*Qin et al., 2012*; *Pogliani et al., 2015*). The single nucleotide polymorphism (SNP) of MTHFR gene (rs1801133, C667T) is correlated with decreased enzyme activity that eventually results in elevated plasma homocysteine levels (*Engbersen et al., 1995*; *Kang et al., 1993*). Hyperhomocysteinemia has been confirmed to be involved in the pathogenesis of stroke, CSVD, various metabolic disorders and so on (*Inamoto et al., 2003*; *Zee et al., 2007*; *Chutinet et al., 2012*; *Qin et al., 2017*; *Nam et al., 2019*; *Piao et al., 2018*; *Kloppenborg et al., 2011*; *Jeon et al., 2014*; *Pavlovic et al., 2011*). In addition, a recent study suggested that MTHFR C677T genotype was associated with CSVD subtype (*Rutten-Jacobs et al., 2016*). However, no previous studies have investigated the mechanism associated with the effect of MTHFR C677T genotype on CSVD susceptibility. In this study, the relationship between SNPs of MTHFR gene (rs1801133, rs1801131, rs2274976, rs4846048, rs4846049, rs13306561 and rs3737964) and CSVD was determined after adjusting for cardiovascular risk factors, and the potential mechanism based on Hcy levels was explored. The aim was to provide useful clues for identifying susceptible populations of CSVD.

## MATERIALS & METHODS

### Participants

A total of 163 consecutive patients with CSVD were collected as the case group in Heze Municipal Hospital between April 2017 and October 2018. In the corresponding period, 326 healthy people were selected as the control group, who were matched to these cases according to age ($\pm 2$ years) and gender at a ratio of 2:1. This study received the permission

of the ethic committee of Heze Municipal Hospital (20160141022), and written informed consent was provided by each participant.

The inclusion criteria of CSVD patients were (1) in accordance with the diagnostic criteria suggested by *Shi & Wardlaw (2016)*; (2) to demonstrate typical neuroimaging changes in the subcortical grey matter and white matter, including white matter hyperintensities (WMHs), prominent perivascular spaces (PVS), cerebral microbleeds (CMBs), atrophy, lacunas and recent small subcortical infarct (*Wardlaw et al., 2013*); (3) complete clinical data; (4) written informed consent. The exclusion criteria included (1) large-artery atherosclerosis; (2) a definite history of subarachnoid hemorrhage or cerebral hemorrhage; (3) acute ischemic stroke caused by cardiogenic embolism.

## Data collection

Demographic data, cardiovascular risk factors and laboratory indexes were collected in each participant. Demographic data included age, gender, height, weight, annual family income, education level and occupation. Cardiovascular risk factors included hypertension, hyperlipemia, diabetes, smoking, drinking and blood pressure. Laboratory indexes included total cholesterol, triacylglycerol (TG), low density lipoprotein cholesterol (LDL-C), high density lipoprotein cholesterol (HDL-C) and fasting blood glucose (FBG).

## SNP genotyping for MTHFR gene

The DNA used in SNP genotyping was extracted from peripheral blood using the salting out method suggested by *Hashemi et al. (2010)*. TaqMan Pre-Designed SNP Genotyping Assays (Applied Biosystems, Carlsbad, USA) was employed to perform SNP genotyping for the MTHFR gene (rs1801133, rs1801131, rs2274976, rs4846048, rs3737966, rs1537515, rs4846049, rs3834044, rs13306561 and rs3737964). The ABI 7500 Fast real-time PCR system (Applied Biosystems, Carlsbad, USA) was employed to perform PCR amplification and allelic discrimination.

## Plasma Hcy assay

The blood sample was collected from the antecubital vein after an overnight fasting in each participant. The blood sample was then separated through centrifugation at 3,000 g for 5 min. The level of Hcy was detected by enzymatic cycling assay using Hcy reagent (Beijing Strong Biotechnologies, Inc, Beijing, China) and the ROCHE Cobas 8000 automatic biochemical analyzer (Roche Ltd., Switzerland).

## Statistical analysis

Haploview software version 4.2 (http://www.broad.mit.edu/mpg/haploview; developed in Mark Daly's laboratory at the Broad Institute) was employed to perform Hardy-Weinberg equilibrium (HWE) test and to calculate allele frequencies and genotype frequencies for all the SNPs (*Barrett, 2009*). Univariate analysis was performed with Student's $t$- test or chi-square test for all variables, including demographic data, vascular risk factors, laboratory indexes and SNP genotyping data. The variables with a $P$ value less than 0.10 were then included in the multivariate analysis, which was used to identify the SNPs associated with CSVD susceptibility through a backward stepwise logistic regression model.

Plasma Hcy levels among different genotypes were compared using ANOVA. All statistical analysis was conducted using the SPSS version 20.0 for Windows (SPSS Inc., USA), and significance was set at $P < 0.05$.

## RESULTS

### Univariate analysis of demographic data, vascular risk factors and laboratory indexes

The case group included 94 males and 69 females, and the control group included 188 males and 138 females. Their average age was (63.28 ± 7.09) years. Univariate analysis demonstrated that age, body mass index (BMI), annual family income, education level, occupation, drinking, hyperlipidaemia, total cholesterol, TG, LDL-C and HDL-C were not statistically different between the case group and the control group, and hypertension, diabetes, smoking, systolic blood pressure (SBP), diastolic blood pressure (DBP) and FBG were statistically different (Table 1).

### SNP analysis

All the SNPs were successfully genotyped in both the case group and the control group. As shown in Table 2, the genotype frequencies of all the SNPs were not statistically different from those evaluated using Hardy-Weinberg equilibrium. Univariate analysis demonstrated that the genotype frequencies of rs1801133 ($\chi^2 = 12.852$, $P = 0.002$) and rs1801131 ($\chi^2 = 6.203$, $P = 0.045$) were statistically different between the case group and the control group, and rs2274976, rs4846048, rs4846049, rs13306561 and rs3737964 were not statistically different (all $P > 0.05$).

### Multivariate analysis

Multivariate analysis was performed to identify the independent association between different genotypes of the MTHFR rs1801133 and rs1801131 and CSVD. The results demonstrated that the polymorphism of rs1801133 was correlated with CSVD after adjusting hypertension, diabetes, smoking, SBP, DBP, FBG, hyperlipidaemia, LDL-C and HDL-C, but rs1801131 was not (Table 3). The MTHFR rs1801133 TT and CT genotype had increased risk for CSVD, and the *OR* was higher in the TT genotype than in the CT genotype (2.307 vs 1.473).

### Plasma Hcy levels

Plasma Hcy levels were compared using ANOVA between the MTHFR rs1801133 TT, CT and CC genotype. The results demonstrated that plasma Hcy levels were highest in the TT genotype, intermediate in the CT genotype, and lowest in the CC genotype (Table 4).

### Joint effect of MTHFR 677C>T (rs1801133) and 1298A>C (rs1801131) on plasma Hcy levels

The combination of genotypes 677TT and 1298CC was abbreviated TTCC, 677TT and 1298AA was abbreviated TTAA, etc. The frequencies of the nine combination genotypes derived from the both polymorphisms were TTAA (35.17%, 172/489), CTAA (24.34%, 119/489), CTAC (12.68%, 62/489), CCAC (11.25%, 55/489), CCAA (8.90%, 44/489),

**Table 1** Univariate analysis results of demographic data, vascular risk factors and laboratory indexes.

| | | Case group (n = 163) | Control group (n = 326) | χ2/t | P |
|---|---|---|---|---|---|
| Age (Years) | | 63.42 ± 7.13 | 63.21 ± 6.98 | 0.309 | 0.804 |
| BMI (Kg/m²) | | 22.18 ± 4.61 | 22.37 ± 4.95 | 0.419 | 0.716 |
| Annual family income (RMB) | <10,000 | 18(11.04%) | 39(11.96%) | | |
| | 10,000–20,000 | 48(29.45%) | 92(28.22%) | 0.138 | 0.933 |
| | >20,000 | 97(59.51%) | 195(59.82%) | | |
| Educational level | Primary school and below | 69(42.33%) | 130(39.88%) | | |
| | Junior high school | 58(35.58%) | 114(34.97%) | 0.596 | 0.742 |
| | Senior high school and above | 36(22.09%) | 82(25.15%) | | |
| Occupation | Farmer | 71(43.56%) | 131(40.18%) | | |
| | Worker | 61(37.42%) | 123(37.73%) | 0.788 | 0.674 |
| | Civil servant/teacher/doctor | 31(19.02%) | 72(22.09%) | | |
| Drinking | Yes | 40(24.54%) | 74(22.70%) | | |
| | No | 123(75.46%) | 252(77.30%) | 0.206 | 0.650 |
| Hyperlipidaemia | Yes | 48(29.45%) | 73(22.39%) | | |
| | No | 115(70.55%) | 253(77.61%) | 2.905 | 0.088 |
| Total cholesterol (mmol/L) | | 4.45 ± 1.53 | 4.30 ± 1.41 | 1.049 | 0.319 |
| TG (mmol/L) | | 1.54 ± 1.22 | 1.42 ± 1.09 | 1.062 | 0.311 |
| LDL-C (mmol/L) | | 2.62 ± 1.34 | 2.39 ± 1.37 | 1.776 | 0.073 |
| HDL-C (mmol/L) | | 1.24 ± 0.48 | 1.33 ± 0.56 | 1.847 | 0.068 |
| Hypertension | Yes | 54(33.13%) | 66(20.25%) | | |
| | No | 109(66.87%) | 260(79.75%) | 9.740 | 0.002 |
| Diabetes | Yes | 50(30.67%) | 64(19.63%) | | |
| | No | 116(69.33%) | 262(80.37%) | 6.797 | 0.009 |
| Smoking | Yes | 49(30.06%) | 65(19.94%) | | |
| | No | 114(69.94%) | 261(80.06%) | 6.228 | 0.013 |
| SBP (mmHg) | | 144.83 ± 13.92 | 138.78 ± 13.88 | 4.535 | <0.001 |
| DBP (mmHg) | | 90.74 ± 9.08 | 86.96 ± 8.64 | 4.41 | <0.001 |
| FBG (mmol/L) | | 6.62 ± 2.85 | 5.59 ± 2.17 | 4.063 | <0.001 |

**Notes.**
BMI, Body mass index; TG, Triacylglycerol; LDL-C, Low density lipoprotein cholesterol; HDL-C, High density lipoprotein cholesterol; SBP, Systolic blood pressure; DBP, Diastolic blood pressure; FBG, Fasting blood glucose.

CCCC (7.16%, 35/489), CTCC (0.20%, 1/489), TTAC (0.20%, 1/489) and TTTC (0, 0%) in all participants.

The distribution of the six common combination genotypes was significantly different between the case group and the control group (Table 5). Plasma Hcy levels were highest in TTAA genotype, and moderate in CTAA and CTAC genotypes, and lowest in CCAC, CCAA and CCCC genotypes. However, the difference was not significant between CTAA and CTAC genotypes and also between CCAC, CCAA and CCCC genotypes (Table 6).

**Table 2  Univariate analysis results of allele and genotype frequency.**

|  |  | Allele frequency n(%) |  | Genotype frequency n(%) |  |  | HWE *P* |
|---|---|---|---|---|---|---|---|
| rs1801133 |  | C | T | CC | CT | TT |  |
|  | Case group* | 125(38.34) | 201(61.66) | 28(17.18) | 69(42.33) | 66(40.49) | >0.05 |
|  | Control group | 325(49.85) | 327(50.15) | 106(32.52) | 113(34.66) | 107(32.82) | >0.05 |
| rs1801131 |  | A | C | AA | AC | CC |  |
|  | Case group* | 282(86.50) | 44(13.50) | 126(77.30) | 30(18.40) | 7(4.30) | >0.05 |
|  | Control group | 516(79.14) | 136(20.86) | 219(67.18) | 78(23.93) | 29(8.90) | >0.05 |
| rs2274976 |  | A | G | AA | AG | GG |  |
|  | Case group | 28(8.59) | 298(91.41) | 2(1.22) | 24(14.72) | 137(84.05) | >0.05 |
|  | Control group | 61(9.36) | 591(90.64) | 6(1.84) | 49(15.03) | 271(83.13) | >0.05 |
| rs4846048 |  | A | G | AA | AG | GG |  |
|  | Case group | 288(88.34) | 38(11.66) | 125(76.69) | 38(23.31) | 0(0) | >0.05 |
|  | Control group | 581(89.11) | 71(10.89) | 255(78.22) | 71(21.78) | 0(0) | >0.05 |
| rs4846049 |  | G | T | GG | GT | TT |  |
|  | Case group | 263(80.67) | 63(19.33) | 106(65.03) | 51(31.29) | 6(3.68) | >0.05 |
|  | Control group | 525(80.52) | 127(19.48) | 213(65.34) | 99(30.37) | 14(4.29) | >0.05 |
| rs13306561 |  | C | T | CC | CT | TT |  |
|  | Case group | 29(8.90) | 297(91.10) | 3(1.84) | 23(14.11) | 137(84.05) | >0.05 |
|  | Control group | 53(8.13) | 599(91.87) | 4(1.23) | 45(13.80) | 277(84.97) | >0.05 |
| rs3737964 |  | A | G | AA | AG | GG |  |
|  | Case group | 37(11.35) | 289(88.65) | 0(0) | 37(22.70) | 126(77.30) | >0.05 |
|  | Control group | 73(11.20) | 579(88.80) | 0(0) | 73(22.39) | 253(77.61) | >0.05 |

**Notes.**
*$P < 0.05$, *vs* allele frequency and genotype frequency of the control group.
HWE, Hardy–Weinberg equilibrium

**Table 3  Independent association between different genotypes of the MTHFR rs1801133 and rs1801131 and CSVD.**

|  | Regression coefficient | Standard error | Wald | *OR* | 95% *CI* | *P* |
|---|---|---|---|---|---|---|
| rs1801133 |  |  | 9.759 |  |  | <0.001 |
| CC |  |  |  | — | — | Ref=1 |
| CT | 0.307 | 0.154 | 7.528 | 1.473 | 1.164–2.258 | 0.003 |
| TT | 0.416 | 0.209 | 11.063 | 2.307 | 1.798–4.141 | <0.001 |
| rs1801131 |  |  | 1.065 |  |  | 0.262 |
| AA |  |  |  | — | — | Ref=1 |
| AC | 0.163 | 0.122 | 0.975 | 0.821 | 0.602–1.479 | 0.323 |
| CC | 0.218 | 0.126 | 1.147 | 0.726 | 0.517–1.368 | 0.218 |

**Notes.**
MTHFR, Methylenetetrahydrofolate reductase; CSVD, Cerebral small vessel disease; *OR*, odds ratio; *CI*, confidence interval.

**Table 4  Plasma Hcy levels in the *MTHFR* rs1801133 CC, CT and TT genotype.**

|  | n | Plasma Hcy levels (pg/ml) |
|---|---|---|
| CC genotype | 134 | $14.96 \pm 4.85$ |
| CT genotype | 182 | $17.04 \pm 5.68^{*}$ |
| TT genotype | 173 | $19.91 \pm 8.73^{*,**}$ |
| F |  | 9.094 |
| P |  | <0.001 |

Notes.
*$P < 0.05$, *vs* CC genotype.
**$P < 0.05$, *vs* CT genotype.

**Table 5  Frequencies of the six common combination genotypes.**

|  | Six common genotype combinations $n$(%) |  |  |  |  |  |
|---|---|---|---|---|---|---|
|  | TTAA | CTAA | CTAC | CCAC | CCAA | CCCC |
| Case group | 66(40.49) | 47(28.83) | 22(13.50) | 6(3.68) | 18(11.04) | 4(2.45) |
| Control group | 106(32.52) | 72(22.09) | 40(12.27) | 49(15.03) | 26(7.98) | 31(9.51) |
| $\chi^2$ |  |  | 25.211 |  |  |  |
| P |  |  | <0.001 |  |  |  |

**Table 6  Plasma Hcy levels of the six common combination genotypes.**

|  | $n$ | Plasma Hcy levels (pg/ml) |
|---|---|---|
| TTAA | 172 | $19.93 \pm 8.71$ |
| CTAA | 119 | $17.76 \pm 5.72^{*}$ |
| CTAC | 62 | $16.32 \pm 5.28^{*}$ |
| CCAC | 55 | $14.75 \pm 4.64^{*,**}$ |
| CCAA | 44 | $15.94 \pm 4.93^{*,**}$ |
| CCCC | 35 | $14.06 \pm 4.49^{*,**,***}$ |
| F |  | 6.275 |
| P |  | 0.019 |

Notes.
*$P < 0.05$ *vs* TTAA genotype.
**$P < 0.05$, *vs* CTAA genotype.
***$P < 0.05$, *vs* CTAC genotype.

# DISCUSSION

As a sulfur-containing amino acid, Hcy is an important intermediate product for the metabolism of methionine. Hcy has an important role in vascular function (*Li et al., 2017*). Elevated Hcy levels can predispose vascular smooth muscle cells and endothelial cells to injury, which leads to activation of coagulation factors, expression of plasminogen activator inhibitor, endothelial proliferation and so on (*Hainsworth et al., 2016*). This further inhibits the expression of thrombomodulin and synthesis of tissue-type plasminogen activator and sulfated heparin, eventually leading to atherogenesis and thrombogenesis through

secretion of inflammatory cytokines, platelet aggregation, endoplasmic reticulum stress, and oxidative stress.

Studies show that Hyperhomocysteinemia (HHcy) is associated with many diseases, including ischemic stroke (IS), CSVD and various metabolic disorders and so on. *Pavlovic et al. (2011)* showed that elevated total Hcy was correlated with clinical status and severity of white matter changes in symptomatic patients with subcortical small vessel disease. HHcy has also been a confirmed independent risk factor for IS (*Wu et al., 2016*; *Boysen et al., 2003*). *Wu et al. (2016)* demonstrated that high Hcy levels were associated with a greater incidence of acute cerebral infarction among patients with carotid artery plaques. *Ji et al. (2015)* reported that high Hcy levels were associated with a poor functional outcome, severe neurological impairment and stroke recurrence in large artery atherosclerosis stroke subtype, which confirmed the atherogenic effect of Hcy. *Lu et al. (2018)* demonstrated that high Hcy levels were correlated with strong plaque enhancement and acute ischemic stroke with adjustment for sex, age, serum creatinine levels and other atherosclerotic risk factors. Several previous cohort studies also showed that high Hcy levels were correlated with increased risk of IS, including the British Regional Heart Study, the Framingham Study and the Northern Manhattan cohort study (*Perry et al., 1995*; *Bostom et al., 1999*; *Sacco et al., 2004*). As a risk factor of atherosclerosis, high Hcy levels are associated with white matter lesions, lacunar infarcts and cognitive impairment. *Kloppenborg et al. (2011)* found that a higher Hcy level was associated with presence of lacunar infarcts and a higher volume of white matter lesions among patients with symptomatic atherosclerotic disease. *Piao et al. (2018)* evaluated the association between Hcy levels and CSVD with the method of meta-analysis. Their results demonstrated that Hcy levels were higher in CSVD patients than in healthy controls. *Nam et al. (2019)* found that serum Hcy levels were associated with the presence of cerebral microbleeds, white matter hyperintensity volume enlarged perivascular spaces in a healthy population. In addition, Hcy levels may be correlated with the susceptibility for NAFLD (*Hu et al., 2016*; *Polyzos et al., 2015*).

MTHFR is a key controlling enzyme involved in the metabolism of Hcy and folate. It is indispensable for the conversion of homocysteine to methionine through catalyzing the transformation of 5, 10-methylenetetrahydrofolate to 5-methyltetrahydrofolate (*Qin et al., 2012*; *Pogliani et al., 2015*). Additionally, it has a role in chromosomal integrity, DNA methylation and maintaining the stability of single- and double-strand DNA (*Robien & Ulrich, 2003*). It is encoded by the MTHFR gene, which is located on chromosome 1p36.3 (*Goyette et al., 1994*). For the MTHFR gene, the cytosine (C) to thymine (T) substitution at position 677 ( rs1801133) in the gene encoding region is the most common SNP. This variation leads to the conversion from alanine to valine at amino acid 222 (*Jadavji et al., 2015*), and is correlated with decrease of thermal stability of MTHFR and subsequent decrease of enzyme activity (*Atadzhanov et al., 2013*; *Ou et al., 2014*). Compared to the CC genotype, the enzyme activity of the CT and TT genotypes is less than 35% and 70%, respectively (*Frosst et al., 1995*). The decreased enzyme activity eventually leads to the elevation of Hcy levels (*Atadzhanov et al., 2013*; *Ou et al., 2014*). In other words, the CT and TT genotypes are correlated with elevated Hcy levels through reducing the activity of MTHFR. *Wang et al. (2018)* found that the SNP of the MTHFR rs1801133 and NAFLD had

a potential synergistic effect on elevated Hcy levels. *Li et al. (2017)* found that the plasma Hcy levels of different genotypes of the MTHFR rs1801133 showed the tendency of the TT genotype >CT genotype >CC genotype. They concluded that a possible synergistic effect of the MTHFR rs1801133 SNP on plasma Hcy levels increased the risk of IS. In addition, Rutten-Jacobs et al. demonstrated that MTHFR C677T genotype was associated with CSVD subtype (*Rutten-Jacobs et al., 2016*).

In this study, the SNP of the MTHFR rs1801133 was correlated with CSVD, and the TT and CT genotypes had increased risk for CSVD compared to the CC genotype. Moreover, the overall response (*OR*) was higher in the TT genotype than in the CT genotype. At the same time, the plasma Hcy levels of different genotypes showed the tendency of the TT genotype >CT genotype >CC genotype. Therefore, the SNP of rs1801133 was correlated with CSVD through elevating Hcy levels. *Ulvik et al. (2007)* demonstrated a strong linkage and functional inference of MTHFR 677C>T and 1298A>C polymorphisms through a large-scale epidemiological investigation. In this study, we further investigated the frequencies of combination genotypes of MTHFR 677C>T and 1298A>C in the case group and the control group. The results showed that the distribution of the six common combination genotypes was significantly different between the case group and the control group. Lastly, we investigated plasma Hcy levels of the six common combination genotypes. However, the results showed that the difference was not significant between CTAA and CTAC genotypes and also between CCAC, CCAA and CCCC genotypes. The reason might be associated with a small sample size.

In this study, the inclusion and exclusion criteria of CSVD patients used in this paper included clinical symptoms, signs and typical neuroimaging changes. Additionally, CSVD patients should have complete clinical data. A ratio of 2:1 was selected for control people: case people with the aim of improving the efficiency of the study through enhancing statistical power. The objective of this study was to evaluate the relationship between SNPs of MTHFR gene and CSVD comprehensively and precisely, and the main limitation was a small sample size. We will investigate the joint effect of MTHFR 677C>T and 1298A>C using a larger sample size in the next step.

## CONCLUSIONS

The SNP of the MTHFR rs1801133 was correlated with CSVD susceptibility through influencing Hcy levels.

### Funding
The authors received no funding for this work.

### Competing Interests
The authors declare there are no competing interests.

## Author Contributions

- Hongyu Yuan and Man Fu performed the experiments, authored or reviewed drafts of the paper, and approved the final draft.
- Xianzhang Yang performed the experiments, prepared figures and/or tables, and approved the final draft.
- Kun Huang analyzed the data, prepared figures and/or tables, and approved the final draft.
- Xiaoyan Ren conceived and designed the experiments, authored or reviewed drafts of the paper, and approved the final draft.

## Human Ethics

The following information was supplied relating to ethical approvals (i.e., approving body and any reference numbers):

This study received the permission of the ethic committee of Heze Municipal Hospital (20160141022), and written informed consent was provided by each participant.

## Data Availability

The raw data are available as a Supplementary File.

## Supplemental Information

Supplemental information for this article can be found online at http://dx.doi.org/10.7717/peerj.8627#supplemental-information.

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
