# Peer review of "Single nucleotide polymorphism of *MTHFR* rs1801133 associated with elevated Hcy levels affects susceptibility to cerebral small vessel disease"

_PeerJ, doi:10.7717/peerj.8627_

## Round 0.1 · original submission · Major Revisions

Critiques of all three reviewers should be carefully addressed and the manuscript should be revised accordingly.

·

Basic reporting

Previous studies have correlated MTHFR SNP causing C677T mutation with cerebral small vascular disease and high plasma homocysteine levels. In this study, authors validate correlations by measuring and comparing blood total homocysteine levels in a case-control study where patients selected for the cohort were screened by CSVD inclusion criteria. Therefore, this study overcomes the limitation of prior studies in direct measurement of tHCY levels as a causal factor for CSVD. The manuscript is written in a clear and scientific manner.However, there are few minor suggestions that would make it even better. They are:

1) Line 60: Shi and Wardlaw, 2016 instead of Shi & Wardlaw.
2) Brief mention of why 2:1 ratio was selected for control people: case people.
3) Name the analyzer used for blood Hcy assay.
4) Please clarify that the values in table 4 are tHCY levels or serum IFN-gamma levels because readout from plasma tHCY enzymatic cycling assay should be tHCY levels. Inflammatory cytokine levels are used as indirect measurement.
5) Present Table 2 in better format. In current format, it is somewhat difficult to read and understand it.

Experimental design

No comment.

Validity of the findings

No comment.

Reviewer 2 ·

Basic reporting

This is a very interesting manuscript investigating the contribution of several SNPs of MTHFR gene to both Hcy levels and susceptibility to cerebral small vessel disease. The authors observed association of rs1801133 with increased risk for CSVD, and the same polymorphism showed a tendency for increased hcy levels. The article is written in good English and professionally structured, but important references have been missed, so that I have some comments to improve its quality (see the next points).

Experimental design

In the introduction (lines 80-81), the authors state "However, the relationship among the MTHFR gene polymorphisms, Hcy, and CSVD has not been investigated". This is not actually true: see for example (a) Increased total homocysteine level is associated with clinical status and severity of white matter changes in symptomatic patients with subcortical small vessel disease. Clin Neurol Neurosurg. 2011 Nov;113(9):711-5. (b) Homocysteine, small-vessel disease, and atherosclerosis: an MRI study of 825 stroke patients. Neurology. 2014 Aug 19;83(8):695-701. (c) Association of MTHFR C677T Genotype With Ischemic Stroke Is Confined to Cerebral Small Vessel Disease Subtype. Stroke. 2016 Mar;47(3):646-51. and many similar articles in the field. The authors must therefore put their research in the context of the available literature in the field, without claiming as original their research hypothesis. Both introduction and discussion should be rewritten in light of the available literature in the field, and particularly their discussion should include and properly discuss previous evidence in the field.

Validity of the findings

1) Univariate analysis demonstrated that the genotype frequencies of rs1801133 (χ2=12.852, P=0.002) and rs1801131 (χ2=6.203, P=0.045) were statistically different between the case group and the control group, but the multivariate analysis only confirmed association for rs181133. It is well known from the literature that rs1801133 and rs1801131are in strong linkage, giving rise to six common genotype combinations and 3 rare combinations (See the article by Ulvik et al. Hum Genet (2007) 121:57–64). In the same article it was also suggested that the effects on circulating Hcy levels are the result of the combined genotypes. I therefore invite the authors to investigate the haplotypes and the combined genotypes for MTHFR 677C>T (rs1801133) and 1298A>C (rs1801131), and clarify if they are different between patients and controls, and their joint effect on Hcy levels.

Additional comments

Please consider my suggestions in order to improve your manuscript and add originality to your investigation

Reviewer 3 ·

Basic reporting

Although the authors report an interesting study, the language used in the manuscript should be revised to ensure better comprehension. The introduction and discussion section lacks cohesion and needs to be revised to avoid possible confusion. Some examples where the language could be improved includes lines 89, 95, 96, 164, 218, 219.

The statement made in lines 64, 65 should be supported by citing appropriate literature.

Experimental design

The authors have mentioned the inclusion and exclusion criteria used for selecting the data they have used for the analysis. These criteria should be discussed in more detail and the authors must explain the reason for the choice of criterion. If available authors must report studies from literature that have used these criteria. This will help validate the results.

Validity of the findings

In the conclusion section, the authors merely, state the result obtained during their analysis. However, no attempt has been made to link the result to potential mechanism of association of the MTHFR rs1801133 SNP with the elevated Hcy levels. Also, the conclusion section is very brief. The authors should address the potential limitation of the study by including the risks/limitations of reaching a conclusion based on analysis of a small data set. How does the size of the data set affect the validity of the findings? Addressing this will make the manuscript more transparent.

Additional comments

In the submitted manuscript titled “Single nucleotide polymorphism of MTHFR rs1801133 associated with elevated Hcy levels affects susceptibility to cerebral small vessel disease” the authors employ standard statistical approaches to find a correlation between SNP of MTHFR rs1801133 and elevated levels of Hcy. Populations with elevated Hcy levels are susceptible to cerebral small vessel disease and the results presented in the manuscript could provide insights into identifying possible markers for the detection of CSVD susceptible populations.
Overall, this is an interesting study, but above mentioned issues/comments need to be addressed before being considered for publication in PeerJ.

---

## Round 0.2 · Major Revisions

Please note that one of the reviewers indicated that only subtle, cosmetic, changes were made to the paper and previous suggestion to conduct additional experiments were not appropriately addressed.

·

Basic reporting

A. The authors haven't addressed comments shared by all reviewers regarding correction of grammar errors in the manuscript. I have tried to make some corrections in the "tracked changes" document that I will forward to the editor to share further.

B. Authors have not provided any table in the revised document for review.

C. The explanation regarding sample size ratio has not been added to the revised manuscript. Kindly add the explanation, including how it statistically improves efficiency of the study.

D. Table 2 still has some formatting issues.

Experimental design

No comment.

Validity of the findings

Discussion and conclusion still need to be revised to make it into a stronger manuscript.

Additional comments

The manuscript looks better after the changes that were earlier suggested have been incorporated. Kindly look into some more suggestions that have been added to further improve your manuscript.

Reviewer 2 ·

Basic reporting

The authors made only cosmetic changes to the manuscript, but failed to address my previous concerns.

Experimental design

The authors included the suggested references.

Validity of the findings

I previously suggested to explore the joint effect of the studied polymorphisms to Hcy levels, the existence of linkage disequilibrium and the distribution of haplotypes. This was not done.

The authors reply was "Thanks for your comments. Your recommendation is very instructive. We will explore their joint effects on CSVD susceptibility and Hcy levels in the next manuscript which we plan to submit to Peer J, too".

They shall do it in the present manuscript.

Additional comments

Please address haplotype distribution and the joint effect contribution of the studied polymorphisms in the present study

---

## Round 0.3 · accepted · Accept

In my view, all the critiques were adequately addressed now and the manuscript was revised accordingly. Therefore, this amended version can be accepted in its present form.